# The HIV 5′ Gag Region Displays a Specific Nucleotide Bias Regulating Viral Splicing and Infectivity

**DOI:** 10.3390/v13060997

**Published:** 2021-05-27

**Authors:** Bastian Grewe, Carolin Vogt, Theresa Horstkötter, Bettina Tippler, Han Xiao, Bianca Müller, Klaus Überla, Ralf Wagner, Benedikt Asbach, Jens Bohne

**Affiliations:** 1Department of Molecular and Medical Virology, Ruhr-University, 44801 Bochum, Germany; BastianGrewe@gmx.de (B.G.); bettina.g.tippler@rub.de (B.T.); han.xiao@uk-erlangen.de (H.X.); bianca.hoffmann@rub.de (B.M.); klaus.ueberla@fau.de (K.Ü.); 2Institute of Virology, Hannover Medical School, 30625 Hannover, Germany; vogt.carolin@mh-hannover.de (C.V.); theresa.horstkoetter@web.de (T.H.); 3Department of Biochemistry, Ruhr-University, 44780 Bochum, Germany; 4Institute of Clinical and Molecular Virology, University Clinics Erlangen, 91054 Erlangen, Germany; 5Institute of Medical Microbiology and Hygiene, University Regensburg, 93053 Regensburg, Germany; ralf.wagner@ukr.de (R.W.); Benedikt.Asbach@klinik.uni-regensburg.de (B.A.); 6Institute of Clinical Microbiology and Hygiene, University Hospital Regensburg, 93053 Regensburg, Germany

**Keywords:** HIV gene expression, U1 snRNP, mRNP code, alternative splicing

## Abstract

Alternative splicing and the expression of intron-containing mRNAs is one hallmark of HIV gene expression. To facilitate the otherwise hampered nuclear export of non-fully processed mRNAs, HIV encodes the Rev protein, which recognizes its intronic response element and fuels the HIV RNAs into the CRM-1-dependent nuclear protein export pathway. Both alternative splicing and Rev-dependency are regulated by the primary HIV RNA sequence. Here, we show that these processes are extremely sensitive to sequence alterations in the 5’coding region of the HIV genomic RNA. Increasing the GC content by insertion of either GFP or silent mutations activates a cryptic splice donor site in *gag*, entirely deregulates the viral splicing pattern, and lowers infectivity. Interestingly, an adaptation of the inserted GFP sequence toward an HIV-like nucleotide bias reversed these phenotypes completely. Of note, the adaptation yielded completely different primary sequences although encoding the same amino acids. Thus, the phenotypes solely depend on the nucleotide composition of the two GFP versions. This is a strong indication of an HIV-specific mRNP code in the 5′ *gag* region wherein the primary RNA sequence bias creates motifs for RNA-binding proteins and controls the fate of the HIV-RNA in terms of viral gene expression and infectivity.

## 1. Introduction

Most retroviruses are faced with the problem of expressing all genes from a single, polycistronic primary transcript (pre-mRNA), which also serves as the genomic RNA to be packaged in newly formed viral particles [1]. In order to generate various translation templates, HIV uses alternative splicing to obtain many mRNAs from a single pre-mRNA [2]. A variety of mechanisms have been described as to how the virus maintains its splicing ratios using HIV reporter constructs [3]. Many of these have served as blueprints to understand the regulation of cellular alternative splicing. In essence, two major features determine splice site usage: splicing enhancers/silencers [2] and dynamic RNA secondary structures [4,5]. Splicing enhancers and silencers are recognized by RNA-binding proteins (RBPs), namely, SR proteins and hnRNPs, respectively [6,7]. Using iCLIP data and bioinformatics approaches, the term splicing code was established to describe and predict splicing events [8,9]. An extension of this concept, named mRNP code, takes into account how the primary RNA sequence dictates all steps of mRNA biogenesis including splicing, processing, translation, and localization based on differential protein recruitment [10,11,12].

Gene expression of HIV is complicated by the use of intron-containing mRNAs for translation [3]. These mRNAs are normally retained in the nucleus [13]. HIV has evolved an export adaptor protein named Rev to overcome this restriction [14]. Rev binds to the Rev-responsive element (RRE) present in all intron-containing viral RNAs and subsequently engages with the cellular protein export factor CRM-1 [15]. Thus, Rev dispatches the incompletely spliced viral RNAs to the cellular protein export pathway, resulting in export of intron-containing viral RNAs.

The sequence features that lead to Rev-dependency have been debated in the last decades [13,16,17,18]. The current view is that a combination of unused splice sites (SS) in the intron-containing RNAs and an HIV-specific sequence bias result in nuclear retention and ultimately degradation of these transcripts [3,19]. Very recently, we could show that spliceosomal factors of the B complex function as retention factors for the unspliced HIV RNA [20]. 

The HIV genome is particularly Adenosin (A)-rich and Cytidin (C)-low, resulting in the utilization of codons that are untypical for human cells [18,21]. By converting the sequence of, for example, HIV *gag* to the human sequence bias, its expression becomes Rev-independent without altering the RRE [18]. This conversion changes the nucleotide bias to Guanosin (G)- and Cytidin (C)-rich sequences. Although this process is usually termed codon-optimization, the enhanced expression levels observed for modified HIV and human herpesvirus 8 (HHV 8) lytic RNAs, which are also characterized by a high AT content, are vastly independent of translation. They stem, rather, from higher RNA levels in the nucleus and cytoplasm [22,23]. Thus, the altered, optimized RNA sequence may bind a different subset of RBPs, resulting in an mRNP code that enhances RNA stability and nuclear export in the absence of Rev [24]. 

In a previous publication, we inserted GFP downstream of the major viral splice donor site 1 (SD1) between the matrix and capsid coding sequence to assess translation from the genomic HIV RNA by flow cytometry [25,26]. In this current work, we moved the GFP insertion site to the 5′ end of *gag,* thereby placing the GFP encoding sequence directly behind either the start codon of gag (matrix) or 45 nts downstream in frame within the matrix coding region. We observed an intriguing, strong Rev-independent GFP expression and a concomitant dramatic change in the HIV splicing pattern, leading to the expression of new HIV RNA isoforms. However, insertion of an adapted GFP gene, which exhibits the HIV sequence bias, restored the RNA processing pattern of the parental wild-type provirus. Interestingly, sequence optimization (mimicking sequence composition of human mRNAs) of the 5′ *gag* sequence in a complete HIV provirus yielded identical results as the respective GFP insertions. A silent mutagenesis screen of the whole HIV genome confirmed the importance of the 5′ region [27]. These findings and the data presented here suggest that non-homologous sequences can yield identical phenotypes solely based on their nucleotide composition and recruited RBPs as a consequence. Finally, the 5′ gag region is critical to maintain the wild-type splicing pattern of HIV and thus its replication fitness.

## 2. Materials and Methods

### 2.1. Proviral Constructs

All proviral Ren constructs are based on the molecular HIV-1 clone NL4.3rev-env [25]. The *env* gene is inactivated by a deletion/frameshift. Rev contains a mutated start and a stop codon at the end of the first exon. All NL4.3Ren-GFP variants contain a stop codon after the GFP ORF. huGFP harbors the F64L mutation in combination with a silent mutation to inactivate a cryptic splice acceptor site. In MAhuGFP/MAhivGFP, the GFP was placed between the matrix and capsid, and in MA45huGFP/MA45hivGFP 45 nucleotides downstream of the matrix ATG. For ∆MAhuGFP/∆MAhivGFP, the GFP sequence was inserted at the start codon of gag. In the huGag construct, the first 717 nt of wt *gag* were substituted by a humanized sequence using the BssHII and SpeI fragment from the plasmid pc_UTR-huGagAB-RRE(NL4-3) [18].

### 2.2. Cells and Transfection

HEK293T cells were grown in DMEM supplemented with 10% fetal calf serum, 1 mM sodium pyruvate, and 1% penicillin-streptomycin. The day before transfection, 4 × 10^5^ HEK293T cells were seeded in a 6-well plate and transfected with 1 µg of the proviral construct, 50 ng tat expression plasmid, 100 ng rev expression plasmid (or empty vector), and 20 ng ß-Galactosidase expression plasmid or 50 ng dsRed as transfection control using either the Nanofectin transfection reagent (Capricorn, Ebsdorfergrund, Germany), ViaFect transfection reagent (Promega, Madison, WI, USA), or Lipofectamine 2000 (Thermo Fisher Scientific, Waltham, MA, USA). Cells were harvested 48 h post transfection. The flow cytometry analysis was performed with the Cytomics FC 500 flow cytometer (Beckman Coulter, Brea, CA, USA). ß-Galactosidase activity was measured using the Mammalian ß-Galactosidase Assay Kit (Thermo Scientific, Waltham, MA, USA) according to the manufacturer’s protocol. For microscopic analyses, 1 × 10^6^ 293T cells were seeded in a 25 cm^2^ plate and transfected via calcium phosphate precipitation with 1 µg of the proviral construct plus 0.5 µg of rev, 0.5 µg tat, and 0.1 µg Gaussia luciferase, which was measured to control for transfection efficiency [25]. The medium was changed after 12 h and two days after transfection, cells were examined under a fluorescence microscope. For titer analysis, the day before transfection, 5 × 10^6^ 293T cells were seeded in a 10 cm plate. Transfections were performed using the calcium phosphate precipitation method with 5 µg of the proviral construct plus 0.5 µg of rev, 0.25 µg tat, 0.5 µg vsvg, and 0.5 µg dsRed as transfection control and 8.25 µg of plasmid DNA. Medium was changed 6 h and the supernatant was harvested 32 and 48 h post transfection. 

### 2.3. Tat-Mediated LTR Activity

LTR-driven expression of the Firefly luciferase reporter gene was quantified with the Firefly/Renilla Dual-Luciferase System (Promega, Madison, WI, USA). Briefly, 3.5 × 10^4^ 293T cells were seeded per well of a white cell culture 96 well plate and transfected one day later by polyethylenimine with 2 ng CMV-Renilla Luc, 25 ng LTR-Firefly Luc, and 12.5 ng of proviral construct or tat or mock [25]. Two days post transfection, luciferase activity was measured and normalized to Renilla as the control for transfection efficiency.

### 2.4. RNA Preparation and Analysis

RNA methods were performed as described previously [28]. A GAPDH-specific probe was prepared by *Eco*RI digestion of a GAPDH plasmid. The RRE probe was generated by digestion of the HIV-gp-RRE plasmid with *Eco*RI/*Bam*HI [29]. The Rev exon 2 probe was generated by a *Bam*HI and *Hind*III digest of pNLCenv. For the huGFP probe, the SF91 GFP vector was digested with *Bam*HI and *Nco*I.

### 2.5. Identification of Splice Donor

For reverse transcription, 10 µg of total RNA was treated with Turbo DNase (Ambion, Austin, TX, USA) and purified with RNeasy Minikit (Qiagen, Hilden, Germany). A total of 800 ng of RNA were reverse transcribed with the QuantiTect Reverse Transcription Kit (Qiagen, Hilden, Germany). For the final PCR, Ren RT1 for (CTC TCG ACG CAG GAC TCG GCT TGC) and Ren RT2 rev (CAG GCC ACG CCT CCC TGG AAA G) primers were used. Reactions were separated on an agarose gel, bands of interest were excised and purified using the QIAquick Gel Extraction Kit, ligated into pCR2.1 vector (TA-Cloning kit; Invitrogen, Waltham, MA, USA), and sequenced using M13 forward and reverse primers. 

### 2.6. Western Blotting

For the detection of HIV capsid (p24), cells were lysed with 1X sodium dodecyl sulfate (SDS) protein lysis buffer (62.5 mM Tris-HCl [pH 6.8], 2% [wt./vol] SDS, 10% glycerol, 50 mM dithiothreitol, 0.01% [wt./vol] bromophenol blue) and boiled for 5 min. Lysates were then analyzed by immunoblotting. p24 was detected with a rabbit anti-p24 HIV antibody in a 1:5000 dilution (kindly provided by H.-G. Kräusslich, University Heidelberg, Heildelberg, Germany). Actin was detected with a mouse monoclonal anti-actin antibody in a 1:1000 dilution (Sigma-Aldrich, St. Louis, MO, USA). Alternatively, cell lysates were prepared using SDS-containing lysis buffer (50 mM Tris-HCl [pH 7.4], 150 mM NaCl, 40 mM NaF, 5 mM EDTA, 5 mM EGTA, 1% [vol/vol] Nonidet P-40, 0.1% [wt./vol] sodium deoxycholate, 0.1% [wt./vol] SDS), debris was removed by centrifugation and proteins were detected with mouse anti-Gag/p24CA (183-H12-5C; NIH AIDS Research and Reference Reagent Program, Los Angeles, CA, USA) and rabbit anti-alpha-tubulin (600-401-880; Rockland, Limerick, PA, USA) antibodies. Alpha-Tubulin was detected on the same blots after stripping with ReBlot Plus Strong Solution (Millipore, Burlington, MA, USA). Detections were carried out by using a standard enhanced chemiluminescence reaction.

### 2.7. Titer Analysis ß-Gal/p24 ELISA

Supernatants from VSV-G pseudotyped wild type Ren or RenhuGag proviruses were harvested 32 and 48 h post transfection and filtered through a 0.22 µm filter. A p24 ELISA (Architect, Abbott, Wiesbaden, Germany) was used to adjust the supernatants before titration on HeLa P4 cells containing integrated LTR-driven β-galactosidase. HeLa P4 cells were maintained in DMEM with 2% Hepes buffer and 4 µg/µL Protaminsulfate. ß-Galactosidase activity was measured after 48 h using the Mammalian ß-Galactosidase Assay Kit (Thermo Scientific, Waltham, MA, USA) according to the manufacturer’s protocol for adherent cells in a microplate.

### 2.8. Bioinformatic Analysis

The analysis of splice sites was performed with Max ent score (http://hollywood.mit.edu/burgelab/maxent/Xmaxentscan_scoreseq.html (accessed on 25 May 2021)). Sequence composition (GC content) was calculated using a customized tool implemented in Object Pascal. Prediction of RNA-binding protein motifs was done using RBPmap (http://rbpmap.technion.ac.il/ (accessed on 25 May 2021)). For comparison of hivGFP with wt Gag, the nucleotide identity over the same length excluding start and stop codons was determined, yielding 31.3% identity. To determine if this value was above random, a simulation of 100 random amino acid sequences of the same length (i.e., 240 aa) was performed. Parameters included the natural amino acid frequency and the HIV codon usage using a sliding-window gene optimization algorithm [30] to obtain a sequence with the same properties and constraints. The resulting HIV-like random nucleotide sequences shared 31.0% identity with hivGFP and wt Gag on average (range 26.1–35.4%). For reanalysis of published deep sequencing data, fasta files (SRR1852860, SRR1852861, and SRR1852881 [31]; SRR5179299 to SRR5179322 [32]; SRR528772 to SRR528881 [33] and ERR1332411 to ERR1332441 [34] SRR8360552 to SRR8360566 and SRR10123175 to SRR10123179 [35]; SRR10973963 to SRR10973971 [20] were obtained. Reads potentially harboring the SDcrCA were identified using a simple custom-made algorithm searching for an exact match with the 15 nt sequence stretch ending with the SDcrCA and being discontinued downstream. Such reads were manually inspected further to delineate the splicing pattern.

## 3. Results

### 3.1. Substitution of Matrix with huGFP Allows Rev-Independent Expression

We cloned a GFP variant named huGFP (see Materials and Methods) between the matrix and capsid (MAhuGFP) as previously described to monitor translation of the genomic RNA by flow cytometry (Figure 1A; [26]). Next, huGFP was inserted further upstream yielding MA45huGFP and retaining the first 45 nucleotides of the matrix frame (i.e., 15 amino acids) to preserve the myristoylation site (Figure 1A). In ∆MAhuGFP, the complete matrix-encoding region was substituted with huGFP (Figure 1A). Stop codons and a deletion of the 5′ part of CA were introduced downstream of all GFP ORFs to prevent expression of a fusion protein with Gag/Pol (Figure 1A). Interestingly, insertion of huGFP at the ATG of the matrix (∆MAhuGFP) resulted in Rev-independent GFP expression, as shown by fluorescence microscopy (Figure 1B). In contrast, the insertion of huGFP between the matrix and capsid (MAhuGFP) requires Rev for expression (Figure 1B). The insertion, which keeps the matrix N-terminus (MA45huGFP), showed lower expression levels in the presence of Rev and some detectable expression without Rev (Figure 1B, middle panel). Thus, moving the insertion site of the huGFP sequence toward the 5′ end of the gag mRNA results in Rev-independent GFP expression.

In a more quantitative approach, we monitored huGFP expression by flow cytometry (Figure 1C). Production of MAhuGFP was strictly Rev-dependent whereas ∆MAhuGFP expression reached similar levels even in the absence of Rev (Figure 1C, left panel). An intermediate Rev-dependency was observed for the MA45huGFP reporter despite low overall GFP expression levels (Figure 1C). Immunoblot analyses of the corresponding lysates gave identical results, proving again the Rev-independent GFP expression for MA45huGFP and ∆MAhuGFP (Figure 1D, left panel, lanes 4,6). MA45GFP yielded lower signals in all methods applied, which argues for protein instability of this particular fusion protein. However, transfected HeLa cells allowed for slightly more MA45huGFP expression, which argues for cell type specific effect in addition (Appendix A).

The insertion of the sequence-optimized huGFP sequence only led to minor overall changes in the nucleotide bias of the entire unspliced HIV gag/pol mRNA (1.4% increase in GC content). However, it massively changed the local sequence composition of the transcript at the 5′ end (roughly 700 nucleotides and 17.8% increase in GC content). To revert these specific local changes, the GC-rich huGFP sequence was replaced by hivGFP (Figure 1A) encoding the same amino acids but utilizing AT-rich codons mimicking the intronic sequences of HIV (Appendix A) [36]. The primary RNA sequence differed in 229 positions (32%) between huGFP and hivGFP. This caused a decrease in GC content by 0.9% for the full-length gag/pol RNA and by 12.2% for the 5’ end (both compared to the wild type), resulting in an even lower GC-content as in the parental Ren construct. Overall, the hivGFP constructs produced much lower GFP levels in all transfections (Figure 1D, right panel). Consequently, the MA45hivGFP expression could neither be detected by FACS nor by western blot (Figure 1C,D; right panels). Importantly, insertion of hivGFP did not induce Rev-independent expression seen for ∆MAhuGFP, but rather restored the wild type expression pattern with Rev-dependent GFP production (Figure 1C, right panel). These results were also corroborated by western blot analyses (Figure 1D, right panel) and point toward the local sequence composition near the 5′ end of the HIV genomic RNA as a determinant of Rev-dependent expression. 

### 3.2. The huGFP Sequence Misregulates HIV-1 Splicing in a Position-Dependent Manner

Since the Rev-independency of GFP-expression seems to increase when the huGFP sequence is moved toward the 5′ end of the gag ORF and thus closer to the major HIV 5′SS, we sought to analyze the splicing pattern of our constructs. Figure 2A depicts the expected regular splicing variants. In order to see differential effects of the huGFP-containing proviruses, we compared them with the parental provirus (Ren) and the hivGFP-bearing versions. No construct expressed Rev to allow for a comparison in the absence and presence of co-transfected Rev. In addition, a Tat expression plasmid was co-transfected to ensure even transactivation levels in cases of disturbed endogenous Tat expression. Moreover, Tat itself has been implicated to affect splicing [37,38] and thus continuous levels are essential to analyze splicing phenotypes here.

Three classes of mRNAs were detectable by northern blot for the parental Ren construct (Figure 2B). The genomic RNA with a size of roughly 9 kb, a class of singly spliced RNA around 4 kb, and the fully spliced RNAs encoding *tat*, *rev*, and *nef* of 2 kb size are displayed (Figure 2B, upper panel, lane 3). Providing Rev *in trans* allowed significant accumulation of singly spliced transcripts and genomic RNA (Figure 2B, upper panel, lanes 2,3). Concomitantly, the amount of fully spliced transcripts decreased (Figure 2B, upper panel, lanes 2,3). Very similar results were seen with the MAhuGFP and MAhivGFP constructs (Figure 2B, upper panel, lanes 4,5 and 10,11). Note that the MAhivGFP lane (Figure 2B, GAPDH probe, lane 10; -Rev) was underloaded. However, in the ∆MAhuGFP and less pronounced in the MA45huGFP, cryptic RNAs of approximately 7–8 kb and 3–4 kb appeared in the absence of Rev (Figure 2B, upper panel, lanes 6,8). In ∆MAhuGFP, the cryptic RNAs dominated the splicing pattern at the expense of the fully spliced RNAs, which were only detectable after prolonged exposure (Figure 2B, upper panel, lane 8). In the presence of Rev, ∆MAhuGFP displayed an almost complete splicing inhibition, resulting in solely genomic RNA (Figure 2B, lane 9). In contrast, the insertion of hivGFP reproduced the wildtype splicing pattern irrespective of the GFP position (Figure 2B, upper panel, lanes 10–15, compared to lanes 2,3), with a tendency to higher levels of the fully spliced transcripts (∆MAhivGFP; Figure 2B, upper panel, lane 14). Thus, the different local primary sequence composition might be responsible for the splicing effects observed. Very similar results were also obtained in northern blots analyzing total RNA from HeLa cells transfected with the huGFP constructs (Appendix A). 

The question remained as to how the cryptic RNAs contribute to Rev-independent expression of huGFP. To analyze this, we re-probed the northern blot with a huGFP specific probe. Interestingly, the huGFP sequence can be detected in the genomic RNAs of every huGFP construct as expected, but also in all cryptic RNAs (Figure 2B, middle panel). Since the GFP sequence is located at the 5′ end, shorter GFP-containing RNAs must be spliced in the 3′ part (Figure 2A). Thus, we used an RRE-specific probe to verify the presence of the *env* intron in which the RRE is embedded. Now the cryptic RNA signals disappeared, meaning that the cryptic 7–8 and 3–4 kb RNAs are spliced in the 3′ part (Figure 2A and Figure 2B lower panel). No signal was observed in the hivGFP lanes due to the differences in the primary GFP sequence (Figure 2B, middle panel).

### 3.3. The Insertion of huGFP Activates a New Cryptic Splice Donor

The unspliced HIV genomic RNA can undergo multiple alternative splicing events yielding more than 40 mRNA isoforms [33,39]. In addition, splicing of HIV-1 occurs in a strict 5′ to 3′ order [40]. To characterize the nature of the cryptic RNAs, RT-PCR using primers binding upstream of the major 5′SS (SD1) and downstream of 3′SS SA7 was performed (Figure 2A) using total RNA from ∆MAhuGFP transfections. We detected a regular splice event from SD1 to SA5 and SD4 to SA7 (Figure 2A, SD1SA5-SD4SA7) normally coding for *nef* [33,39], which was underrepresented in the northern blot (Figure 2B, upper panel, lane 8). More importantly, we also detected a new HIV splice event originating from a new cryptic splice donor site in the capsid (CA) gene (SDcrCA) to SA3 and from SD4 to SA7, corresponding to the ~3 kb band in the northern blot (Figure 2B). The same cryptic SD was identified in a study while introducing silent mutations throughout the HIV genome [27], thus confirming our results. However, the mechanism of activation might be different. The SDcr is located 227 nt downstream of the huGFP stop codon (Figure 2A). Hence, the insertion of huGFP but not hivGFP leads to the activation of this cryptic SD site in CA. The SDcrCA has a Max ent score of 7.3 compared to 10.1 for SD1. Usage of this cryptic site in combination with regular downstream 3′SS generates seemingly fully spliced mRNAs lacking all introns but containing the huGFP ORF. Except for the major SD1, such transcripts would not contain unused splice sites and might consequently function as huGFP mRNA (Figure 2A, SDcrCASA5-SD4SA7). To determine the usage of the cryptic SDcrCA in the wild type context, we re-evaluated deep-sequencing data from previous studies. Emery et al. [32] amplified singly and doubly spliced RNAs. Among the close to 30 million reads, we identified 11 potential hits. Nine of these represented splice events were consistent with our observations (i.e., SDcrCAA5-SD3SA7 (seven times) and SDcrCAA2-SD3A4b-D4A7 (two times)), while the others may represent artifacts. In the study by Sherrill-Mix et al. [31], we found 25 hits among nearly 700 million reads. However, only one showed splicing from SDcrCA to an unknown acceptor approx. 700 nt downstream in Pol. The other hits involved acceptor sequences upstream of SDcrCA, thus most likely representing artifacts or unusual trans-splicing events. Sequencing of HIV RNAs by Nanopore technology obtained several potential hits involving the SDcrCA, though all of them turned out to be artifacts upon manual inspection [41]. No hits were found in other sequencing data assessed [20,33,34]. Thus, splicing from SDcrCA represents an extremely rare, but possible event in the wild type context. However, an altered nucleotide composition 200 nts upstream makes SDcrCA the dominant splice donor site in the absence of Rev. 

### 3.4. The Insertion of huGFP Decreased Tat-Mediated LTR Activity

In our initial experiments, Tat was supplied *in-trans* to ensure even transcriptional enhancement. The marked reduction of fully spliced HIV RNAs in ∆MAhuGFP (Figure 2B, upper panel, lane 8 and 9) prompted us to ask if *tat* mRNA levels and Tat protein quantity were also reduced. We investigated the capability of our proviral constructs to activate an HIV-1 LTR promoter-driven Firefly luciferase reporter as a surrogate for Tat quantity. Shifting the huGFP sequence to the 5′ end of the genomic RNA reduced Tat-dependent LTR-induction from 20- to 3.5-fold (Figure 3A). In contrast, LTR-induction was constantly high after cotransfection of all hivGFP containing proviruses (Figure 3A). This argues for profoundly decreased Tat expression levels in co-transfections with ∆MAhuGFP (Figure 3A). 

In addition, we repeated the northern blot analyses of the huGFP proviruses from cells without Tat co-transfection. Here, the ∆MAhuGFP construct produced lower levels of all RNA species compared to the parental provirus (Figure 3B, upper panel, lanes 7, 8). However, insertions further downstream as in MA45huGFP and MAhuGFP produced similar RNA levels independent of Tat co-transfection (Figure 3B, upper panel, lanes 3–6). Importantly, the splicing pattern remained identical, irrespective of Tat being present or not (compare Figure 2B and Figure 3B), arguing for a splicing rather than a transcriptional phenotype. Thus, the 5′ insertion of the huGFP sequence inhibits the major SD1 and reduces the amount of fully spliced viral transcripts including the *tat* mRNA, which in turn leads to lower transactivation from the LTR-promoter. 

### 3.5. Humanizing 5′ HIV Gag Yields the Identical Phenotype as a 5′ huGFP Insertion

Decreased splicing and generation of cryptically spliced transcripts could be a specific characteristic of the optimized huGFP sequence. In order to test if a local G/C-rich nucleotide bias would, in general, induce the same effects, we substituted the first 717 nt (comparable to the insertion of 723 nts huGFP) of wt *gag* by a humanized sequence. This adaptation yielded a provirus encoding the identical Gag amino acid sequence, but with an 11% increase in the GC content of the *gag* ORF (huGag) and an overall increase by only 0.7% (Figure 1A). 

Northern blot analyses revealed that insertion of the humanized Gag sequence at the very 5′ end of *gag* affected the viral splicing pattern in exactly the same way as huGFP in ∆MAhuGFP, although the primary sequences completely differed and encoded the non-related proteins GFP and Gag (Figure 4A, compare lanes 7 and 8 vs. 9 and 10). Comparison with the parental provirus demonstrated that huGag induced the expression of cryptically spliced RNAs in the absence of Rev and reduced the generation of fully spliced transcripts (Figure 4A, lanes 1 and 2 vs. 9 and 10). This alteration of the splicing pattern in the huGag provirus should also generate a distinct Gag/Pol protein expression pattern. Immunoblot analyses using antibodies directed against CA demonstrated a clearly Rev-dependent expression pattern in the parental provirus (Figure 4B, lanes 1,2). In contrast, the huGag construct demonstrated Rev-independent expression of several protein variants carrying the epitope recognized by the monoclonal α-p24 antibody (Figure 4B, lanes 3,4). The band migrating just below the Pr55 (Gag) (marked with an asterisk; compare lane 2 to lane 3) might stem from the translation of cryptically spliced RNAs (SDcrCASA5-SD4SA7) of roughly 3 kb in size (Figure 2A) giving rise to a translation product of 53.5 kDa. The higher migrating band of around 70 kDa (Figure 4B; marked with an asterisk) could be translated from an RNA after splicing from SDcrCA to the normally infrequently used SA6 in *env* [42] and continuous translation into the *env* frame. Interestingly, this band was also observed in the silent mutagenesis screen [27]. Taken together, the local sequence alterations at the 5′ end lead to Rev-independent protein expression as observed with the huGFP insertion (Figure 1). No fully processed CA protein could be detected (Figure 4B, lane 3), instead, an intermediate cleavage product appears, which is not readily explained by the splicing pattern in the absence of Rev (Figure 4A, lane 9). In the presence of Rev, the aberrant Gag band almost completely disappeared. The pattern is indistinguishable from the parental construct (Figure 4B, compare lanes 2 and 4). This is consistent with the RNA analysis showing that Rev leads to the accumulation of genomic RNA only (Figure 4A, lane 10).

Finally, we tested whether the huGag provirus also displayed reduced infectivity. Supernatants of transfections using the parental Ren construct or the huGag provirus in the presence of Rev and Tat and VSV-G envelope expressing plasmids were collected. To rule out differences in particle amount, we normalized for p24 content and titrated the supernatants on HelaP4 cells harboring an integrated LTR-driven, Tat-dependent β-galactosidase. As shown in Figure 4C, transduction with huGag leads to very low β-galactosidase expression levels. Based on our northern blot data (Figure 4A, lane 9 and 10) we assume that the reduction of fully spliced RNAs by the huGag sequence, which in turn leads to reduced Tat levels and LTR-induction (compare Figure 3A), is also effective for integrated proviruses.

In summary, the humanization of the first approx. 700 nts of the *gag* coding sequence reproduced the effects of a huGFP insertion. Even though the underlying sequences of huGFP and huGag only share 36% identity (slightly above random), the effects on the HIV splicing pattern and Rev-independent protein expression are similar. 

## 4. Discussion

In general, the primary mRNA sequence holds more information than just the codons for translation. RNA is folded into secondary structures and recognized by RBPs. Thus, underneath the genetic code, an mRNP code exists. The primary sequence of the HIV genomic RNA is characterized by an AT-rich nucleotide bias (e.g., in the *gag* coding region). This bias constitutes an over-representation of binding motifs for certain RBPs. This HIV-specific mRNP code is crucial to maintain a favorable ratio of spliced vs. unspliced RNAs (Figure 2). However, sequences in the pol region tolerate substitutions of Adenosines without affecting viral replication [43]. It seems that, in particular, the coding region of the matrix is crucial for the activity of the major 5′ splice site and Rev-dependency. This satellization on 5′ sequences as decision-making determinants was previously observed [44]. Alterations introduced either by silent mutations (huGag, Figure 4 or [27]) or insertion of a gene carrying a different sequence bias (e.g., huGFP) changes the local nucleotide composition and in essence, the binding sites for RBPs and/or the secondary structure. These changes have drastic consequences for the splicing pattern of HIV, leading to new, cryptically spliced RNAs. These are (i) Rev-independent; (ii) generated from a new 5′ splice site (SDcrCA; Figure 2); and (iii) constitute a new class of singly spliced RNAs with removal of the *env* intron, which does not occur in wt. HIV [40]. The usage of cryptic splice sites can also be enforced by mutations, increasing the stability of the secondary structure holding the major HIV 5′SS [45]. 

The Rev-independent expression of our cryptically spliced transcripts can be explained by the removal of all intronic sequences and thus, inhibitory splicing complexes [20]. This occurs by functional inactivation of SD1 and simultaneous splicing from the cryptic SD and additional splicing events in the 3′ region. However, these RNAs should be subjected to nonsense-mediated RNA decay (NMD), since they are spliced in their 3′UTR with the GFP stop codon being more than 50 nts upstream of the splice junction. However, this was not the case. Perhaps the NMD system is overwhelmed in our transient transfection situation. In addition, 293T cells exhibit inefficient NMD [46]. However, we observed the same NMD-independent GFP expression in HeLa cells (Supplementary Figure 2). Therefore, an alternative explanation could be that NMD is not effective on the HIV-derived RNAs, as has been suggested for Rous-sarcoma virus RNAs [47,48].

The suppression of the major SD1 and simultaneous activation of the cryptic splice site is probably caused by the recruitment of a different set of RBPs to huGFP or the humanized *gag* sequence. Since sequence or codon usage adaption is independent of translation of HIV [22], we believe that an altered mRNP code is responsible for the observed effects, as we could also show in α-herpesviruses [23]. We performed an in silico analysis for the abundance of putative binding motifs for hnRNPs or SR proteins in the wild type or the optimized huGag sequence. We observed an increase in the number of motifs for SR proteins in the humanized sequence and a decrease for the hnRNP motifs (Figure 4D). Regarding the splicing phenotype, it was shown that sequences in HIV matrix encode a splicing enhancer for SD1 (Figure 4E and Appendix A; [49,50]). This splicing enhancer is characterized by three hnRNP and two SR protein binding motifs as determined by our in-silico prediction. However, the humanized Gag sequence as well as the huGFP insertion delete this enhancer and display a strong bias to SR protein binding motifs. As a result, splicing at the new cryptic 5′SS occurs (Figure 4E). SR proteins are splicing inhibitory if they bind in introns (Figure 4E; [51]). Interestingly, the insertion of huGFP between the matrix and capsid, which is only 400 nts downstream, did not change the splicing pattern as previously described (Figure 1 and Figure 2; [26]). Our data marked the region downstream of the matrix ATG as very sensitive to changes in the local nucleotide composition. This is further corroborated by reversion of the phenotype if the GFP sequence bias was shifted back to an HIV-like composition (Figure 1 and Figure 2). Although the 5′ *gag* sequence and the HIV-like GFP do not share any homology above random (Appendix A and Methods for detailed analysis), the outcome is similar. In addition, the HIV-like GFP sequence shows high-scoring binding motifs for hnRNPs. This means that solely the sequence bias and the resulting mRNP code determine the fate of the HIV RNAs. An alternative explanation would be the involvement of an RNA secondary structure. However, the hivGFP sequence is so unrelated to the natural matrix sequence that formation of a similar RNA structure is highly unlikely. 

As above-mentioned, humanization of the HIV wild type matrix region exactly reproduced the effects of the huGFP insertion (Figure 4). Thus, similar to the relationship between hivGFP and the wild type HIV sequence, primarily, the sequence bias determines the fate of the HIV RNAs. Our observations are in full agreement with the work of the Bieniasz laboratory [27]. Here, activation of the same cryptic splice donor was observed as well as distorted Gag/Pol protein production in the absence of Rev and a decrease in viral titer. Both GFP/Nef as in [27] and Tat as in our findings (Figure 4C) depend on the expression of fully spliced RNAs, which are severely reduced by the sequence alterations in the 5′ gag region (Figure 4A). The addition of Rev restores expression of the genomic RNA from proviral constructs containing codon-optimized sequences in the 5′ *gag* region. We speculate that Rev removes HIV full-length RNA quickly from the nucleus, leaving not enough time for (cryptic) splicing to occur [13,52].

The sequence composition of the HIV genomic RNA not only regulates splicing and Rev-dependency, but also viral packaging [53] and moreover affects recognition by intrinsic restriction factors [54]. This work, together with the data presented from the whole provirus silent mutagenesis [27], points to a specific viral mRNP code for HIV that controls the splicing pattern, viral gene expression, and infectivity.

## Figures and Tables

**Figure 1 viruses-13-00997-f001:**
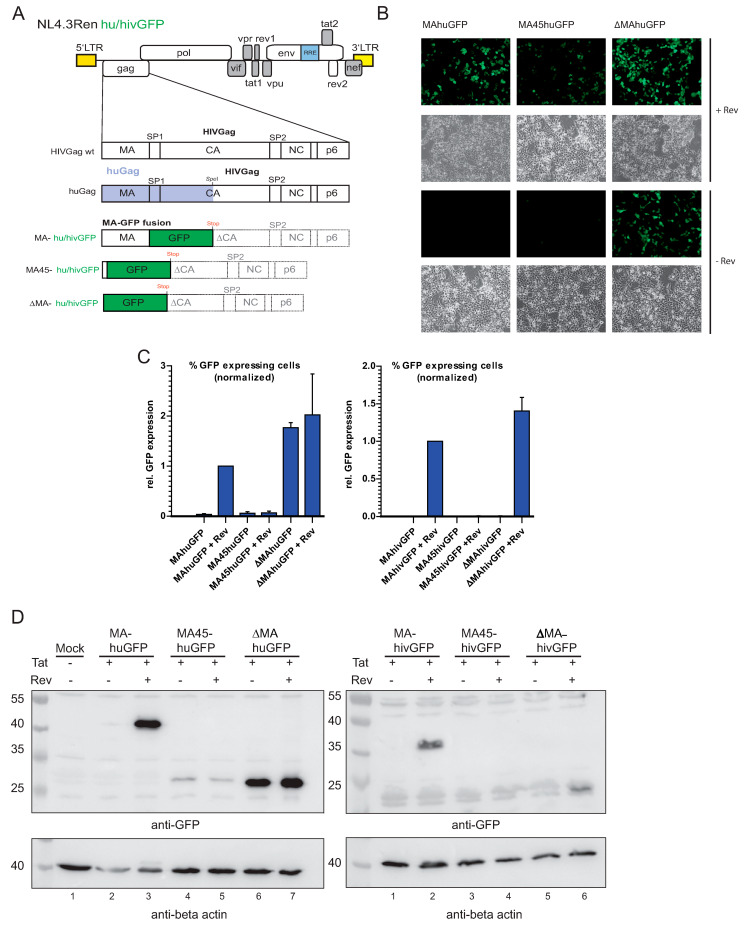
Substitution of the matrix-encoding sequence with huGFP allows Rev-independent expression. (**A**) Depiction of the proviral HIV-1 construct NL4.3Ren (upper part). The LTRs are marked as yellow rectangles. The Rev response element is highlighted in blue. Below a magnification of the *gag* ORF is shown. Matrix (MA) as well as capsid (CA) and nucleocapsid (NC) are interspaced by the spacer peptides SP1 and 2. At the very C-terminus, the p6 protein is depicted. Next, a chimeric Gag is shown consisting of a sequence optimized part (light violet, up to the SpeI restriction site) and wild type sequences. At the bottom, the three different positions of GFP insertions (humanized (hu) or adapted to HIV nucleotide bias (hiv)) are illustrated: between the matrix and capsid (MAhuGFP/MAhivGFP), after the first 45 nt (i.e., 15 amino acids) of matrix (MA45huGFP/ MA45hivGFP), and at the ATG of matrix (∆MAhuGFP/∆MAhivGFP). The GFP ORF is terminated by a STOP codon followed by a small deletion in CA, leaving the rest of Gag untranslated (light gray dashed rectangles). (**B**) Light and fluorescence microscopy pictures of HEK 293T cells transfected with the different proviral constructs with or without a Rev expression plasmid. (**C**) Flow cytometry analysis of relative GFP expression levels of HEK 293T cells transfected with the indicated constructs. The percentage of GFP expressing cells was normalized to the overall transfection efficiency using β-galactosidase expression as a control for transfection efficiency (*n* = 3). Error bars represent SD. Both the huGFP and hivGFP expression levels were separately normalized to the MAhuGFP + Rev or MAhivGFP + Rev control, respectively. (**D**) Western blot using protein extracts from the indicated co-transfections. Detection was performed with an anti-eGFP antibody. Actin served as a loading control. A size marker in kDa is indicated on the left.

**Figure 2 viruses-13-00997-f002:**
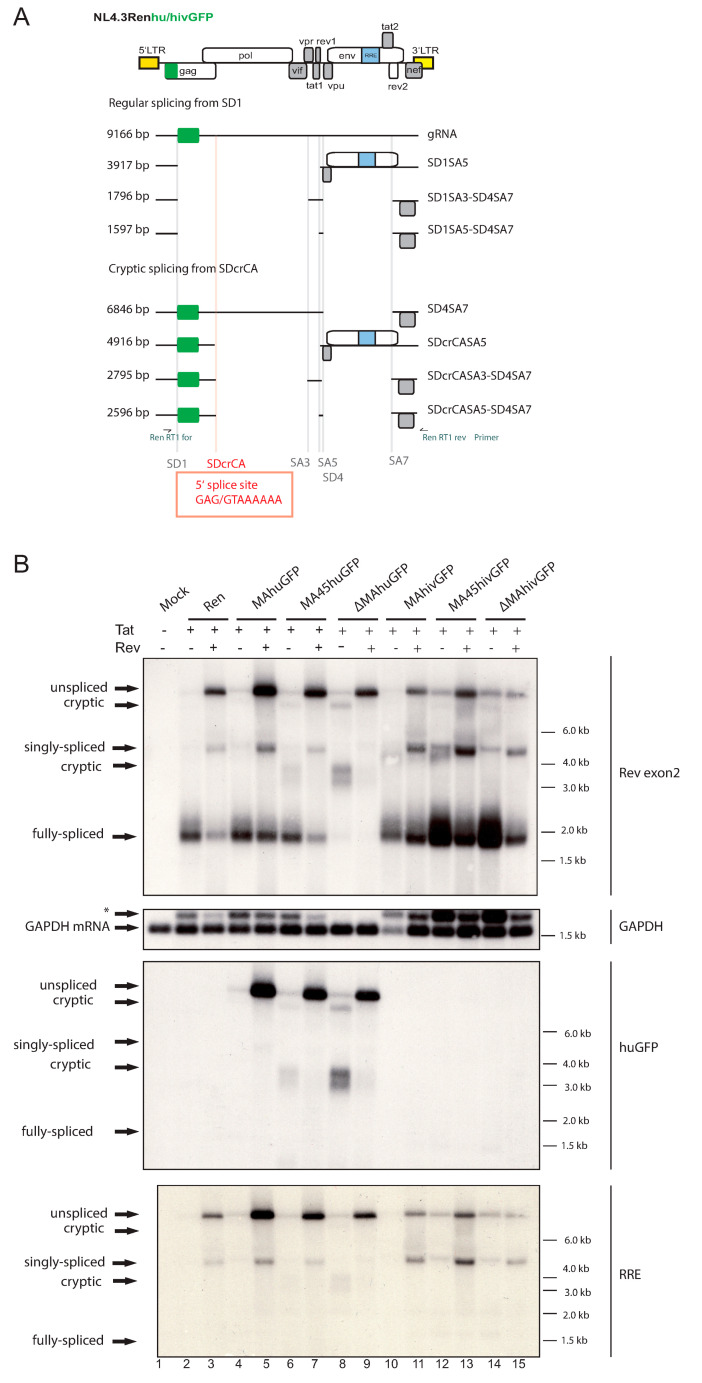
huGFP misregulates HIV-1 splicing in a position-dependent manner. (**A**) Depiction of the proviral genome (top panel). The predicted splice variants for the ∆MAhuGFP construct are shown below. In the upper part, splicing from the regular splice donor 1 (SD1) is depicted and in the lower part, splicing events from the newly identified cryptic splice donor (SDcrCA) are shown. The sizes of the RNA variants are given on the left side and the possible combinations of splice donor and acceptor sites are depicted on the right margin. At the very bottom, the splice sites are shown with their position on the proviral genome. The primers used for RT-PCR are depicted at the bottom of the panel as half arrows. (**B**) Northern blot of total RNA from HEK 293T cells transfected with the indicated constructs. Membranes were rehybridized with different probes (named on the right-hand side) specific for Rev exon2, huGFP, RRE, and GAPDH, which served as loading control. The different splice variant classes are indicated on the left, sizes are given on the right, and an asterisk marks residual signal from the previous hybridization.

**Figure 3 viruses-13-00997-f003:**
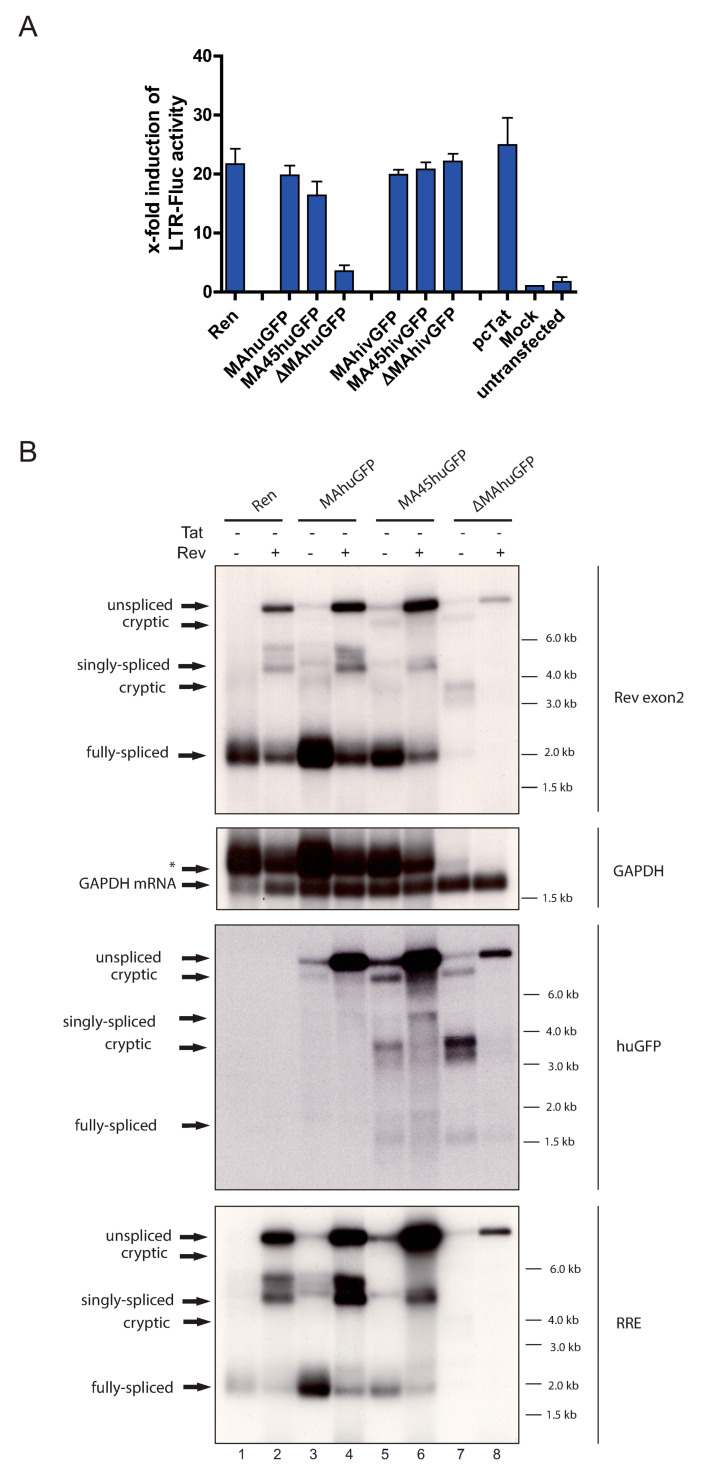
The insertion of huGFP decreased Tat-mediated LTR activity. (**A**) LTR promoter-driven expression of Firefly luciferase was measured after co-transfection of the indicated proviral constructs and was normalized to CMV-Renilla luciferase expression, which served as the transfection control. All experiments were done in the absence of Rev. *n* = 4. Error bars represent SD. (**B**) Northern blot of total RNA from HEK 293T cells transfected with the indicated proviral constructs in the absence of Tat. Membranes were rehybridized with different probes specific for Rev exon 2, huGFP, RRE, and GAPDH, which served as the loading control. The different splice variants are indicated on the left, sizes are given on the right, and residual signal from the previous hybridization is marked by an asterisk.

**Figure 4 viruses-13-00997-f004:**
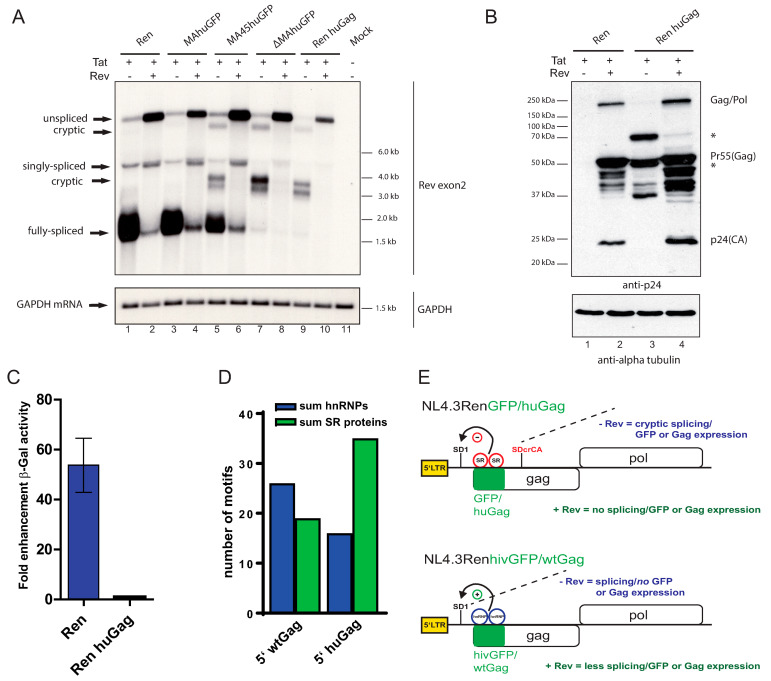
Humanizing 5′ HIV *gag* sequences yields the identical phenotype as a huGFP insertion. (**A**) Northern blot of total RNA from HEK 293T cells transfected with the indicated proviral constructs with huGFP insertions compared to the splicing pattern of a construct with a humanized 5′HIV *gag* sequence (Ren huGag). Membranes were hybridized with a probe specific for Rev exon 2 and rehybridized with a GAPDH probe (both indicated on the right), which served as the loading control. The different splice variants are indicated on the left and the RNA sizes on the right. (**B**) Western blot using protein extracts from the indicated co-transfections. Detection was performed with an anti-p24 antibody. Alpha tubulin served as a loading control. The newly detected Gag protein isoforms are marked with an asterisk. A size marker in kDa is indicated on the left. The asterisks mark unknown Gag isoforms. (**C**) Titer analysis using supernatants containing VSV-G pseudotyped wildtype Ren- or Ren huGag-derived particles. A p24 ELISA was used to adjust the supernatants before titration on HeLa P4 cells containing integrated β-galactosidase under the control of an HIV-1 long terminal repeat. Data from three independent experiments with supernatants from two different harvesting time points per experiment are shown (*n* = 6). Error bars represent SD. (**D**) Analysis of putative binding motifs for hnRNP or SR proteins by RBPmap (http://rbpmap.technion.ac.il/ (accessed on 05/25/2021)). The first 700 nts of either wild type HIV *gag* or the humanized version were analyzed. (**E**) Model depicting the influence of the mRNP code on the activity of the major SD1. In the case of huGFP or the humanized Gag sequence, SR proteins preferentially bind and suppress SD1, leading to activation of the cryptic SD (SDcrCA, in red). In the case of the HIV-like GFP or the wild type Gag sequence, hnRNPs are recruited that activate the SD1.

## Data Availability

Data are contained within the article or Appendix A.

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
