# Peer review of "The HIV 5′ Gag Region Displays a Specific Nucleotide Bias Regulating Viral Splicing and Infectivity"

_viruses, 2021, doi:10.3390/v13060997_

Round 1

Reviewer 1 Report

In this truly interesting manuscript, the authors describe important aspects of the mechanism by which the HIV Rev protein facilitates and regulates splicing-dependent RNA translation during the viral life cycle.

Although using the highly artificial system of recombinant viral genomes, where the authors insert into the gag-pol sequence entirely unrelated genes, the approach uses quite elegant ways of demonstrating the sole sequence-dependence (GC content, HIV-like nucleotide bias) rather than gene-related information (Gag) to be responsible for the Rev-dependence of expression.

Overall, the information is very interesting to the specialists in the field, maybe a bit less for the general reader.

This feature, termed by the authors "HIV-specific mRNP code" could be very relevant for HIV research and for a better understanding of the molecular mechanisms of HIV replication.

The manuscript is well-written, experimental design and description are quite clear, and the conclusions have a solid support by the presented experimental results.

Only a few minor points remain open that should be addressed by the authors:

  • line 56/57: maybe the authors may emphasise the (otherwise obvious) fact that this manipulation in gag does NOT affect the RRE sequence (in env)
  • line 71: ...optimization...yielded identical results as the GFP insertion" - could be rephrased to make it clear that  both gag versions behave like the respective GFP insertions (?)
  • in the discussion, line 335, the authors mention the use of 293T cells (with inefficient NMD) - as this appears to be important, the authors are encouraged to include an experiment also with another cell line for this point!

minor:

word missing in line 251: "...of these represented splice events (were) consistent with our observations,..."

Out of curiosity: Interesting to note that the authors find that Ca-Phosphate transfection (line 97) of the full-length constructs worked superior to the other transfection agents(?)

Author Response

Reviewer 1:

Minor point (1): line 56/57: maybe the authors may emphasise the (otherwise obvious) fact that this manipulation in gag does NOT affect the RRE sequence (in env).

This is a good suggestion and the respective sentence has been revised.

Minor point (2): line 71: ...optimization...yielded identical results as the GFP insertion" - could be rephrased to make it clear that both gag versions behave like the respective GFP insertions (?)

We rephrased the sentence to make this point clearer.

Minor point (3): in the discussion, line 335, the authors mention the use of 293T cells (with inefficient NMD) - as this appears to be important, the authors are encouraged to include an experiment also with another cell line for this point!

We now included HeLa cells in the new supplementary figure 2. Interestingly, as in HEK 293T cells, GFP expression from the cryptic RNAs has been observed. The respective sentence in the discussion has been rephrased. We now strongly suggest that HIV RNAs by some means escape NMD. 

Minor point (4): word missing in line 253: "...of these represented splice events (were) consistent with our observations,..."

The sentence has been corrected.

Minor point (5): Out of curiosity: Interesting to note that the authors find that Ca-Phosphate transfection (line 97) of the full-length constructs worked superior to the other transfection agents(?)

In our experience, 293T cells display the highest efficiency using the traditional Ca-Phosphate precipitation method. However, the pH of the HBS solution is critical.

Reviewer 2 Report

In this manuscript, Grewe et al. investigated the mechanism by which alternative splicing and Rev-dependency are regulated by the primary RNA sequence of HIV-1. The authors found that the increased GC content activates a cryptic splice donor site in gag, deregulates the viral splicing pattern, and impairs the virus infectivity. They also indicated that an adaptation of the inserted GFP sequence towards an HIV-like nucleotide bias reversed these phenotypes completely.

Although their findings are new and add useful knowledge to the HIV research field, and the experiments themselves seem to be neatly performed, several important experiments and points should be performed and/or improved to verify their conclusions. Specific comments by the reviewer are described below.

(Comments)

  • The authors used HEK293T cells throughout the experiments. They should show that the same or similar results are obtained using at least one different cell type. Although the authors mentioned the results using HeLa cells in the text as for the experiments of figure 4, no descriptions were observed for the other experiments in the manuscript. The authors should demonstrate those results using HeLa cells as well.
  • To clearly demonstrate the effect of Rev on both the splicing pattern and nuclear export of the viral RNAs, the northern blot analyses should be performed in not only the whole cell extracts but also the cytoplasm fractions and nuclear fractions.
  • To demonstrate reproducibility of the experiments, the authors should indicate the number of repeated experiments throughout the experiments. In addition, if the data are quantitative ones, they should be expressed as mean+/-SD.
  • Figure 1: the authors should add Tat (-)/ Rev (-) as a negative control in the panels B and C as they did in the panel D.
  • Figure 4C: the authors should show the production levels and viral RNA contents/certain amounts of virions of the VSG-G pseudotyped Ren and Ren hugag HIV-1. In addition, the reviewer asks the authors to demonstrate that what step(s) of the early phase of HIV-1 are impaired in the Ren hugag HIV-1.
  • Author contributions: the reviewer thinks that B.H. is supposed to be B.M.

Author Response

Reviewer 2:

Major point (1): The authors used HEK293T cells throughout the experiments. They should show that the same or similar results are obtained using at least one different cell type. Although the authors mentioned the results using HeLa cells in the text as for the experiments of figure 4, no descriptions were observed for the other experiments in the manuscript. The authors should demonstrate those results using HeLa cells as well.

The reviewer is correct and we added a complete data set consisting of an immunofluorescence analysis as well as flow cytometry and a Northern blot. This is now presented in Supplementary Figure 2.

Major point (2): To clearly demonstrate the effect of Rev on both the splicing pattern and nuclear export of the viral RNAs, the northern blot analyses should be performed in not only the whole cell extracts but also the cytoplasm fractions and nuclear fractions.

Although we agree with the reviewer on the importance of RNA fractionation, we emphasize that the detectable GFP expression or the expression of the cryptic Gag variants can only stem from RNAs that underwent nuclear export. Since the Rev-independent expression is the focus of this manuscript, we do not think that a fractionation experiment would add significantly to the paper.

Major point (3): To demonstrate reproducibility of the experiments, the authors should indicate the number of repeated experiments throughout the experiments. In addition, if the data are quantitative ones, they should be expressed as mean+/-SD.

We corrected this throughout the figure legends. 

Major point (4): Figure 1: the authors should add Tat (-)/ Rev (-) as a negative control in the panels B and C as they did in the panel D.

Tat/Rev minus control in Fig. 1D are simple mock 293T cells. In our view adding them to panel 1B does not alter the Rev-independent phenotype. In Fig. 1C mock 293T cells (Tat/Rev minus) serve as controls for the gating strategy. These original flow cytometry data could be provided upon request.

Major point (5): Figure 4C: the authors should show the production levels and viral RNA contents/certain amounts of virions of the VSG-G pseudotyped Ren and Ren hugag HIV-1. In addition, the reviewer asks the authors to demonstrate that what step(s) of the early phase of HIV-1 are impaired in the Ren hugag HIV-1.

The supernatants used in Fig. 4C originate from transient transfections using gag/pol expression plasmids in addition. However, we detect less genomic RNA in the huGag transfections. In contrast, the immunoblot in Fig. 4B shows even higher levels of huGag. With the normalization on the amount of p24, we tried to account for possible differences. Nevertheless, the reviewer is correct that we did not assess the RNA content of the virions. However, the difference in genomic RNA expression and thus possibly less packaging cannot account for the stark difference observed in beta-Gal expression in Fig. 4C.

Major point (6): In addition, the reviewer asks the authors to demonstrate that what step(s) of the early phase of HIV-1 are impaired in the Ren hugag HIV-1.

We show that Tat expression is severely affected (Fig. 3B) and therefore included a Tat-expression plasmid in all transfections. Since the huGFP insertion and the partial Gag optimization resulted in exactly the same splicing pattern, we extrapolate that Tat expression is also similarly reduced. This means the early phase of Ren huGag is characterized by less transactivation and lower levels of transcription from the LTR. In the absence of Rev the RNAs are cryptically spliced and yield a new class of Rev-independent HIV RNAs (Fig. 4A).

In addition, the optimized gag sequence could have an impact on reverse transcription itself and would lead to less integrated provirues. However, we believe that answering this question goes beyond the scope of this manuscript. 

Minor point: In The reviewer thinks that B. H. is supposed to be B. M.

Bianca Müller (neé Hoffmann) kept her email address directing to her maiden name “Hoffmann”.

Round 2

Reviewer 2 Report

I  have confirmed that the revised manuscript has been sufficiently improved
to warrant publication in this journal.